# Boosted Sparse and Low-Rank Tensor Regression

**Lifang He**
Weill Cornell Medicine
lifanghescut@gmail.com

**Kun Chen**[*]
University of Connecticut
kun.chen@uconn.edu

**Wanwan Xu**
University of Connecticut
wanwan.xu@uconn.edu

**Jiayu Zhou**
Michigan State Universtiy
dearjiayu@gmail.com

**Fei Wang**
Weill Cornell Medicine
few2001@med.cornell.edu

## Abstract

We propose a sparse and low-rank tensor regression model to relate a univariate outcome to a feature tensor, in which each unit-rank tensor from the CP decomposition of the coefficient tensor is assumed to be sparse. This structure is both parsimonious and highly interpretable, as it implies that the outcome is related to the features through a few distinct pathways, each of which may only involve subsets of feature dimensions. We take a divide-and-conquer strategy to simplify the task into a set of sparse unit-rank tensor regression problems. To make the computation efficient and scalable, for the unit-rank tensor regression, we propose a stagewise estimation procedure to efficiently trace out its entire solution path. We show that as the step size goes to zero, the stagewise solution paths converge exactly to those of the corresponding regularized regression. The superior performance of our approach is demonstrated on various real-world and synthetic examples.

## 1 Introduction

Regression analysis is commonly used for modeling the relationship between a predictor vector $\mathbf{x} \in \mathbb{R}^I$ and a scalar response $y$. Typically a good regression model can achieve two goals: accurate prediction on future response and parsimonious interpretation of the dependence structure between $y$ and $\mathbf{x}$ [Hastie et al., 2009]. As a general setup, it fits $M$ training samples $\{(\mathbf{x}^m, y^m)\}_{m=1}^M$ via minimizing a regularized loss, i.e., a loss $L(\cdot)$ plus a regularization term $\Omega(\cdot)$, as follows

$$\min_w \frac{1}{M} \sum\nolimits_{m=1}^M L(\langle \mathbf{x}^m, \mathbf{w} \rangle, y^m) + \lambda \Omega(\mathbf{w}), \tag{1}$$

where $\mathbf{w} \in \mathbb{R}^I$ is the regression coefficient vector, $\langle \cdot, \cdot \rangle$ is the standard Euclidean inner product, and $\lambda > 0$ is the regularization parameter. For example, the sum of squared loss with $\ell_1$-norm regularization leads to the celebrated LASSO approach [Tibshirani, 1996], which performs sparse estimation of $\mathbf{w}$ and thus has implicit feature selection embedded therein.

In many modern real-world applications, the predictors/features are represented more naturally as higher-order tensors, such as videos and Magnetic Resonance Imaging (MRI) scans. In this case, if we want to predict a response variable for each tensor, a naive approach is to perform linear regression on the vectorized data (e.g., by stretching the tensor element by element). However, it completely ignores the multidimensional structure of the tensor data, such as the spatial coherence of the voxels. This motivates the tensor regression framework [Yu and Liu, 2016, Zhou et al., 2013], which treats each observation as a tensor $\mathcal{X}$ and learns a tensor coefficient $\mathcal{W}$ via regularized model fitting:

$$\min_W \frac{1}{M} \sum\nolimits_{m=1}^M L(\langle \mathcal{X}^m, \mathcal{W} \rangle, y^m) + \lambda \Omega(\mathcal{W}). \tag{2}$$

---

[*]Corresponding Author

When $\ell_1$-norm regularization is used, this formulation is essentially equivalent to (1) via vectorization. To effectively exploit the structural information of $\mathcal{X}^m$, we can impose a low-rank constraint on $\mathcal{W}$ for Problem (2). Some authors achieved this by fixing the CANDECOMP/PARAFAC (CP) rank of $\mathcal{W}$ as a priori. For example, Su et al. [2012] assumed $\mathcal{W}$ to be rank-1. Since rank-1 constraint is too restrictive, Guo et al. [2012] and Zhou et al. [2013] imposed a rank-$R$ constraint in a tensor decomposition model, but none of the above methods considered adding sparsity constraint as well to enhance the model interpretability. Wang et al. [2014] imposed a restrictive rank-1 constraint on $\mathcal{W}$ and also applied an elastic net regularization [Zou and Hastie, 2005]. Tan et al. [2012] imposed a rank-$R$ constraint and applied $\ell_1$-norm regularization to factor matrices to also promote sparsity in $\mathcal{W}$. Signoretto et al. [2014] applied the trace norm (nuclear norm) for low-rank estimation of $\mathcal{W}$, and Song and Lu [2017] imposed a combination of trace norm and $\ell_1$-norm on $\mathcal{W}$. Bengua et al. [2017] showed that the trace norm may not be appropriate for capturing the global correlation of a tensor as it provides only the mean of the correlation between a single mode (rather than a few modes) and the rest of the tensor. In all the above sparse and low-rank tensor models, the sparsity is imposed on $\mathcal{W}$ itself, which, does not necessarily lead to the sparsity on the decomposed matrices.

In this paper, we propose a sparse and low-rank tensor regression model in which the unit-rank tensors from the CP decomposition of the coefficient tensor are assumed to be sparse. This structure is both parsimonious and highly interpretable, as it implies that the outcome is related to the features through a few distinct pathways, each of which may only involve subsets of feature dimensions. We take a divide-and-conquer strategy to simplify the task into a set of sparse unit-rank tensor factorization/regression problems (SURF) in the form of

$$\min_{W} \frac{1}{M} \sum_{m=1}^{M} (y^m - \langle \mathcal{X}^m, \mathcal{W} \rangle)^2 + \lambda \|\mathcal{W}\|_1, \quad \text{s.t.} \quad \text{rank}(\mathcal{W}) \le 1,$$

To make the solution process efficient for the SURF problem, we propose a boosted/stagewise estimation procedure to efficiently trace out its entire solution path. We show that as the step size goes to zero, the stagewise solution paths converge exactly to those of the corresponding regularized regression. The effectiveness and efficiency of our proposed approach is demonstrated on various real-world datasets as well as under various simulation setups.

## 2 Preliminaries on Tensors

We start with a brief review of some necessary preliminaries on tensors, and more details can be found in [Kolda and Bader, 2009]. We denote scalars by lowercase letters, e.g., $a$; vectors by boldfaced lowercase letters, e.g., $\mathbf{a}$; matrices by boldface uppercase letters, e.g., $\mathbf{A}$; and tensors by calligraphic letters, e.g., $\mathcal{A}$. We denote their entries by $a_i$, $a_{i,j}$, $a_{i,j,k}$, etc., depending on the number of dimensions. Indices are denoted by lowercase letters spanning the range from 1 to the uppercase letter of the index, e.g., $n = 1, \cdots, N$. Each entry of an $N$th-order tensor $\mathcal{A} \in \mathbb{R}^{I_1 \times \cdots \times I_N}$ is indexed by $N$ indices $\{i_n\}_{n=1}^{N}$, and each $i_n$ indexes the $n$-mode of $\mathcal{A}$. Specifically, $-n$ denotes every mode except $n$.

**Definition 1** (Inner Product). *The inner product of two tensors $\mathcal{A}, \mathcal{B} \in \mathbb{R}^{I_1 \times \cdots \times I_N}$ is the sum of the products of their entries, defined as $\langle \mathcal{A}, \mathcal{B} \rangle = \sum_{i_1=1}^{I_1} \cdots \sum_{i_1=1}^{I_N} a_{i_1,\cdots,i_N} b_{i_1,\cdots,i_N}$.*

It follows immediately that the Frobenius norm of $\mathcal{A}$ is defined as $\|\mathcal{A}\|_F = \sqrt{\langle \mathcal{A}, \mathcal{A} \rangle}$. The $\ell_1$ norm of a tensor is defined as $\|\mathcal{A}\|_1 = \sum_{i_1=1}^{I_1} \cdots \sum_{i_N=1}^{I_N} |\mathcal{A}_{i_1,\cdots,i_N}|$.

**Definition 2** (Tensor Product). *Let $\mathbf{a}^{(n)} \in \mathbb{R}^{I_n}$ be a length-$I_n$ vector for each $n = 1, 2, \cdots, N$. The tensor product of $\{\mathbf{a}^{(n)}\}$, denoted by $\mathcal{A} = \mathbf{a}^{(1)} \circ \cdots \circ \mathbf{a}^{(N)}$, is an $(I_1 \times \cdots \times I_N)$-tensor of which the entries are given by $\mathcal{A}_{i_1,\cdots,i_N} = a_{i_1}^{(1)} \cdots a_{i_N}^{(N)}$. We call $\mathcal{A}$ a rank-1 tensor or a unit-rank tensor.*

**Definition 3** (CP Decomposition). *Every tensor $\mathcal{A} \in \mathbb{R}^{I_1 \times \cdots \times I_N}$ can be decomposed as a weighted sum of rank-1 tensors with a suitably large $R$ as*

$$\mathcal{A} = \sum_{r=1}^{R} \sigma_r \cdot \mathbf{a}_r^{(1)} \circ \cdots \circ \mathbf{a}_r^{(N)}, \tag{3}$$

*where $\sigma_r \in \mathbb{R}$, $\mathbf{a}_r^{(n)} \in \mathbb{R}^I$ and $\|\mathbf{a}_r^{(n)}\|_2 = 1$.*

**Definition 4** (Tensor Rank). *The tensor rank of $\mathcal{A}$, denoted by $\text{rank}(\mathcal{A})$, is the smallest number $R$ such that the equality (3) holds.*

**Definition 5** ($n$-mode Product). *The $n$-mode product of a tensor $\mathcal{A} \in \mathbb{R}^{I_1 \times \cdots \times I_N}$ by a vector $\mathbf{u} \in \mathbb{R}^{I_n}$, denoted by $\mathcal{A} \times_n \mathbf{u}$, is an $(I_1 \times \cdots \times I_{n-1} \times I_{n+1} \cdots \times I_N)$-tensor of which the entries are given by $(\mathcal{A} \times_n \mathbf{u})_{i_1,\ldots,i_{n-1}i_{n+1},\ldots,i_N} = \sum_{i_n=1}^{I_n} a_{i_1,\ldots,i_N} u_{i_n}$.*

## 3 Sparse and Low-Rank Tensor Regression

### 3.1 Model Formulation

For an $N$th-order predictor tensor $\mathcal{X}^m \in \mathbb{R}^{I_1 \times \cdots \times I_N}$ and a scalar response $y^m$, $m = 1, \cdots, M$, we consider the regression model of the form

$$y^m = \langle \mathcal{X}^m, \mathcal{W} \rangle + \varepsilon^m \tag{4}$$

where $\mathcal{W} \in \mathbb{R}^{I_1 \times \cdots \times I_N}$ is an $N$th-order coefficient tensor, and $\varepsilon^m$ is a random error term of zero mean. Without loss of generality, the intercept is set to zero by centering the response and standardizing the predictors as $\sum_{m=1}^{M} y^m = 0$; $\sum_{m=1}^{M} \mathcal{X}^m_{i_1, \cdots, i_N} = 0$ and $\sum_{m=1}^{M} (\mathcal{X}^m_{i_1, \cdots, i_N})^2 / M = 1$ for $i_n = 1, \cdots, I_n$. Our goal is to estimate $\mathcal{W}$ with $M$ i.i.d. observations $\{(\mathcal{X}^m, y^m)\}_{m=1}^{M}$.

To reduce the complexity of the model and leverage the structural information in $\mathcal{X}^m$, we assume that the coefficient tensor $\mathcal{W}$ to be both low-rank and sparse. Specifically, we assume $\mathcal{W}$ can be decomposed via CP decomposition as $\mathcal{W} = \sum_{r=1}^{R} \mathcal{W}_r = \sum_{r=1}^{R} \sigma_r \mathbf{w}_r^{(1)} \circ \cdots \circ \mathbf{w}_r^{(N)}$, where each rank-1 tensor $\mathcal{W}_r$ is possibly sparse, or equivalently, the vectors in its representation $\mathbf{w}_r^{(n)}$, $n = 1, \ldots, N$, are possibly sparse, for all $r = 1, \ldots, R$.

Here we impose sparsity on the rank-1 components from CP decomposition – rather than on $\mathcal{W}$ itself [Chen et al., 2012, Tan et al., 2012]. This adaption can be more beneficial in multiple ways: 1) It integrates a finer sparsity structure into the CP decomposition, which enables a direct control of component-wise sparsity; 2) It leads to an appealing model interpretation and feature grouping: the outcome is related to the features through a few distinct pathways, each of which may only involve subsets of feature dimensions; 3) It leads to a more flexible and parsimonious model as it requires less number of parameters to recover the within-decomposition sparsity of a tensor than existing methods which impose sparsity on the tensor itself, thus makes the model generalizability better.

A straightforward way of conducting model estimation is to solve the following optimization problem:

$$\min_{\sigma_r, w_r^{(n)}} \frac{1}{M} \sum_{m=1}^{M} (y^m - \langle \mathcal{X}^m, \sum_{r=1}^{R} \sigma_r \mathbf{w}_r^{(1)} \circ \cdots \circ \mathbf{w}_r^{(N)} \rangle)^2 + \sum_{r=1}^{R} \sum_{n=1}^{N} \lambda_{r,n} \|\mathbf{w}_r^{(n)}\|_1,$$

$$\text{s.t.} \quad \|\mathbf{w}_r^{(n)}\|_1 = 1, \ n = 1, \cdots, N, \ r = 1, \cdots, R. \tag{5}$$

where $\lambda_{r,n}$, $r = 1, \cdots, R$, $n = 1, \cdots, N$, are regularization parameters. This problem is very difficult to solve because: 1) The CP rank $R$ needs to be pre-specified; 2) As the CP decomposition may not be unique, the pursue of its within-decomposition sparsity is highly non-convex and the problem may suffer from parameter identifiability issues [Mishra et al., 2017]; 3) The estimation may involve many regularization parameters, for which the tuning becomes very costly.

### 3.2 Divide-and-Conquer: Sequential Pursue for Sparse Tensor Decomposition

We propose a divide-and-conquer strategy to recover the sparse CP decomposition. Our approach is based on the sequential extraction method (a.k.a. deflation) [Phan et al., 2015, Mishra et al., 2017], which seeks a unit-rank tensor at a time and then deflates to find further unit-rank tensors from the residuals. This has been proved to be a rapid and effective method of partitioning and concentrating tensor decomposition. Specifically, we sequentially solve the following sparse unit-rank problem:

$$\widehat{\mathcal{W}}_r = \min_{\mathcal{W}_r} \frac{1}{M} \sum_{m=1}^{M} (y_r^m - \langle \mathcal{X}^m, \mathcal{W}_r \rangle)^2 + \lambda_r \|\mathcal{W}_r\|_1 + \alpha \|\mathcal{W}_r\|_F^2, \quad \text{s.t.} \quad \text{rank}(\mathcal{W}_r) \leq 1. \tag{6}$$

where $r$ is the sequential number of the unit-rank terms and $y_r^m$ is the current residue of response with

$$y_r^m := \begin{cases} y^m, & \text{if } r = 1 \\ y_{r-1}^m - \langle \mathcal{X}^m, \widehat{\mathcal{W}}_{r-1} \rangle, & \text{otherwise,} \end{cases}$$

where $\widehat{\mathcal{W}}_{r-1}$ is the estimated unit-rank tensor in the $(r-1)$-th step, with tuning done by, e.g., cross validation. The final estimator is then obtained as $\widehat{\mathcal{W}}(R) = \sum_{r=1}^{R} \widehat{\mathcal{W}}_r$. Here for improving the convexity of the problem and its numerical stability we have used the elastic net [Zou and Hastie, 2005] penalty form instead of LASSO, which is critical to ensure the convergence of the optimization solution. The accuracy of the solution can be controlled simply by adjusting the values of $\lambda_r$ and $\alpha$. Since we mainly focus on sparse estimation, we fix $\alpha > 0$ as a small constant in numerical studies.

As each $\mathcal{W}_r$ is of unit rank, it can be decomposed as $\mathcal{W}_r = \sigma_r \mathbf{w}_r^{(1)} \circ \cdots \circ \mathbf{w}_r^{(N)}$ with $\sigma_r \geq 0$ and $\|\mathbf{w}_r^{(n)}\|_1 = 1, n = 1, \cdots, N$. It is then clear that $\|\mathcal{W}_r\|_1 = \|\hat{\sigma}_r \hat{\mathbf{w}}_r^{(1)} \circ \cdots \circ \hat{\mathbf{w}}_r^{(N)}\|_1 = \sigma_r \prod_{n=1}^{N} \|\mathbf{w}_r^{(n)}\|_1$. That is, the sparsity of a unit-rank tensor directly leads to the sparsity of its components. This allows us to kill multiple birds with one stone: by simply pursuing the element-wise sparsity of the unit-rank coefficient tensor with only one tuning parameter $\lambda$, solving (6) can produce a set of sparse factor coefficients $\hat{\mathbf{w}}_r^{(n)}$ for $n = 1, \cdots, N$ simultaneously.

With this sequential pursue strategy, the general problem boils down to a set of sparse unit-rank estimation problems, for which we develop a novel stagewise/boosting algorithm.

## 4 Fast Stagewise Unit-Rank Tensor Factorization (SURF)

For simplicity, we drop the index $r$ and write the generic form of the problem in (6) as

$$\widehat{\mathcal{W}} = \min_{W} \frac{1}{M} \sum_{m=1}^{M} (y^m - \langle \mathcal{X}^m, \mathcal{W} \rangle)^2 + \lambda \|\mathcal{W}\|_1 + \alpha \|\mathcal{W}\|_F^2, \quad \text{s.t.} \quad \text{rank}(\mathcal{W}) \leq 1. \quad (7)$$

Let $\mathcal{W} = \sigma \mathbf{w}^{(1)} \circ \cdots \circ \mathbf{w}^{(N)}$, where $\sigma \geq 0$, $\|\mathbf{w}^{(n)}\|_1 = 1$, and the factors $\mathbf{w}^{(n)}, n = 1, \cdots, N$ are identifiable up to sign flipping. Let $\mathbf{y} = [y^1, \cdots, y^M] \in \mathbb{R}^M$, and $\mathcal{X} = [\mathcal{X}^1, \cdots, \mathcal{X}^M] \in \mathbb{R}^{I_1 \times \cdots \times I_N \times M}$. Then (7) can be reformulated as

$$\min_{\sigma, w^{(n)}} \frac{1}{M} \|\mathbf{y} - \sigma \mathcal{X} \times_1 \mathbf{w}^{(1)} \times_2 \cdots \times_N \mathbf{w}^{(N)}\|_2^2 + \lambda \sigma \prod_{n=1}^{N} \|\mathbf{w}^{(n)}\|_1 + \alpha \sigma^2 \prod_{n=1}^{N} \|\mathbf{w}^{(n)}\|_2^2$$

$$\text{s.t.} \quad \sigma \geq 0, \ \|\mathbf{w}^{(n)}\|_1 = 1, \ n = 1, \cdots, N. \quad (8)$$

Before diving into the stagewise/boosting algorithm, we first consider an alternating convex search (ACS) approach [Chen et al., 2012, Minasian et al., 2014] which appears to be natural for solving (7) with any fixed tuning parameter. Specifically, we alternately optimize with respect to a block of variables $(\sigma, \mathbf{w}^{(n)})$ with others fixed. For each block $(\sigma, \mathbf{w}^{(n)})$, the relevant constraints are $\sigma \geq 0$ and $\|\mathbf{w}^{(n)}\|_1 = 1$, but the objective function in (8) is a function of $(\sigma, \mathbf{w}^{(n)})$ only through their product $\sigma \mathbf{w}^{(n)}$. So both constraints are avoided when optimizing with respect to $\sigma \mathbf{w}^{(n)}$. Let $\hat{\mathbf{w}}^{(n)} = \sigma \mathbf{w}^{(n)}$ and $\mathbf{Z}^{(-n)} = \mathcal{X} \times_1 \mathbf{w}^{(1)} \times_2 \cdots \times_{n-1} \mathbf{w}^{(n-1)} \times_{n+1} \cdots \times_N \mathbf{w}^{(N)}$, the subproblem boils down to

$$\min_{\hat{w}^{(n)}} \frac{1}{M} \|\mathbf{y}^{\mathrm{T}} - \mathbf{Z}^{(-n)\mathrm{T}} \hat{\mathbf{w}}^{(n)}\|_2^2 + \alpha \beta^{(-n)} \|\hat{\mathbf{w}}^{(n)}\|_2^2 + \lambda \|\hat{\mathbf{w}}^{(n)}\|_1, \quad (9)$$

where $\beta^{(-n)} = \prod_{l \neq n} \|\mathbf{w}^{(l)}\|_2^2$. Once we obtain the solution $\hat{\mathbf{w}}^{(n)}$, we can set $\sigma = \|\hat{\mathbf{w}}^{(n)}\|_1$ and $\mathbf{w}^{(n)} = \hat{\mathbf{w}}^{(n)}/\sigma$ to satisfy the constraints whenever $\hat{\mathbf{w}}^{(n)} \neq \mathbf{0}$. If $\hat{\mathbf{w}}^{(n)} = \mathbf{0}$, $\mathbf{w}^{(n)}$ is no longer identifiable, we then set $\sigma = 0$ and terminate the algorithm. Note that when $\alpha > 0$, each subproblem is strongly convex and the generated solution sequence is uniformly bounded, we can show that ACS can converge to a coordinate-wise minimum point of (8) with properly chosen initial value [Mishra et al., 2017]. The optimization procedure then needs to be repeated to a grid of tuning parameter values for obtaining the solution paths of parameters and locating the optimal sparse solution along the paths.

Inspired by the biconvex structure of (7) and the stagewise algorithm for LASSO [Zhao and Yu, 2007, Vaughan et al., 2017], we develop a fast stagewise unit-rank tensor factorization (SURF) algorithm to trace out the entire solution paths of (7) in a single run. The main idea of a stagewise procedure is to build a model from scratch, gradually increasing the model complexity in a sequence of simple learning steps. For instance, in stagewise estimation for linear regression, a forward step searches for the best predictor to explain the current residual and updates its coefficient by a small increment, and a backward step, on the contrary, may decrease the coefficient of a selected predictor to correct for greedy selection whenever necessary. Due to the biconvex or multi-convex structure of our objective function, it turns out that efficient stagewise estimation remains possible: the only catch is that, when we determine which coefficient to update at each iteration, we always get $N$ competing proposals from $N$ different tensor modes, rather than just one proposal in case of LASSO.

To simplify the notations, the objective function (9) is re-arranged into a standard LASSO as

$$\min_{\hat{w}^{(n)}} \frac{1}{M} \|\hat{\mathbf{y}} - \hat{\mathbf{Z}}^{(-n)} \hat{\mathbf{w}}^{(n)}\|_2^2 + \lambda \|\hat{\mathbf{w}}^{(n)}\|_1 \quad (10)$$

---

**Algorithm 1** Fast Stagewise Unit-Rank Tensor Factorization (SURF)

---

**Input:** Training data $\mathcal{D}$, a small stepsize $\epsilon > 0$ and a small tolerance parameter $\xi$ [2]

**Output:** Solution paths of $(\sigma, \{\mathbf{w}^{(n)}\})$.

1: *Initialization*: take a forward step with $(\{\hat{i}_1, \cdots, \hat{i}_N\}, \hat{s}) = \underset{\{i_1, \cdots, i_N\}, s=\pm\epsilon}{arg\,min} J(s\mathbf{1}_{i_1}, \mathbf{1}_{i_2}, \cdots, \mathbf{1}_{i_N})$, and

$$\sigma_0 = \epsilon, \; \mathbf{w}_0^{(1)} = sign(\hat{s})\mathbf{1}_{\hat{i}_n}, \; \mathbf{w}_0^{(n)} = \mathbf{1}_{\hat{i}_n}(n \neq 1), \; \lambda_0 = (J(\{\mathbf{0}\}) - J(\sigma_0, \{\mathbf{w}_0^{(n)}\}))/\epsilon. \quad (11)$$

    Set the active index sets $I_0^{(n)} = \{\hat{i}_n\}$ for $n = 1, \cdots, N; t = 0$.

2: **repeat**

3:    *Backward step*:

$$(\hat{n}, \hat{i}_{\hat{n}}) := \arg\underset{n, i_n \in I_t^{(n)}}{min} J(\hat{\mathbf{w}}_t^{(n)} + s_{i_n}\mathbf{1}_{i_n}), \; \text{ where } \; s_{i_n} = -sign(\hat{w}_{t i_n}^{(n)})\epsilon. \quad (12)$$

    **if** $\Gamma(\hat{\mathbf{w}}_t^{(\hat{n})} + s_{\hat{i}_{\hat{n}}}\mathbf{1}_{\hat{i}_{\hat{n}}}; \lambda_t) - \Gamma(\hat{\mathbf{w}}_t^{(\hat{n})}; \lambda_t) \leq -\xi$, **then**

$$\sigma_{t+1} = \|\hat{\mathbf{w}}_t^{(\hat{n})} + s_{\hat{i}_{\hat{n}}}\mathbf{1}_{\hat{i}_{\hat{n}}}\|_1, \; \mathbf{w}_{t+1}^{(\hat{n})} = (\hat{\mathbf{w}}_t^{(\hat{n})} + s_{\hat{i}_{\hat{n}}}\mathbf{1}_{\hat{i}_{\hat{n}}})/\sigma_{t+1}, \; \mathbf{w}_{t+1}^{(-\hat{n})} = \mathbf{w}_t^{(-\hat{n})},$$

$$\lambda_{t+1} = \lambda_t, \quad I_{t+1}^{(n)} := \begin{cases} I_t^{(\hat{n})} \setminus \{\hat{i}_{\hat{n}}\}, & \text{if } w_{(t+1)\hat{i}_{\hat{n}}}^{(\hat{n})} = 0 \\ I_t^{(n)}, & \text{otherwise.} \end{cases}$$

4:    **else** *Forward step*:

$$(\hat{n}, \hat{i}_{\hat{n}}, \hat{s}_{\hat{i}_{\hat{n}}}) := \arg\underset{n, i_n, s=\pm\epsilon}{min} J(\hat{\mathbf{w}}_t^{(n)} + s_{i_n}\mathbf{1}_{i_n}), \quad (13)$$

$$\sigma_{t+1} = \|\hat{\mathbf{w}}_t^{(\hat{n})} + \hat{s}_{\hat{i}_{\hat{n}}}\mathbf{1}_{\hat{i}_{\hat{n}}}\|_1, \; \mathbf{w}_{t+1}^{(\hat{n})} = (\hat{\mathbf{w}}_t^{(\hat{n})} + \hat{s}_{\hat{i}_{\hat{n}}}\mathbf{1}_{\hat{i}_{\hat{n}}})/\sigma_{t+1}, \; \mathbf{w}_{t+1}^{(-\hat{n})} = \mathbf{w}_t^{(-\hat{n})},$$

$$\lambda_{t+1} = \min[\lambda_t, \frac{J(\sigma_t, \{\mathbf{w}_t^{(n)}\}) - J(\sigma_{t+1}, \{\mathbf{w}_{t+1}^{(n)}\}) - \xi}{\Omega(\sigma_{t+1}, \{\mathbf{w}_{t+1}^{(n)}\}) - \Omega(\sigma_t, \{\mathbf{w}_t^{(n)}\})}], \quad I_{t+1}^{(n)} := \begin{cases} I_t^{(n)} \cup \{\hat{i}_n\}, & \text{if } n = \hat{n} \\ I_t^{(n)}, & \text{otherwise.} \end{cases}$$

5:    Set $t = t + 1$.

6: **until** $\lambda_t \leq 0$

---

using the augmented data $\hat{\mathbf{y}} = (\mathbf{y}, \mathbf{0})^{\mathrm{T}}$ and $\hat{\mathbf{Z}}^{(-n)} = (\mathbf{Z}^{(-n)}, \sqrt{\alpha\beta^{(-n)}M}\mathbf{I})^{\mathrm{T}}$ , where $\mathbf{I}$ is the identity matrix of size $I_n$. We write the objective function (10) by

$$\Gamma(\hat{\mathbf{w}}^{(n)}; \lambda) = J(\hat{\mathbf{w}}^{(n)}) + \lambda\Omega(\hat{\mathbf{w}}^{(n)}).$$

We use $(\sigma, \{\mathbf{w}^{(n)}\})$ to denote all the variables if necessary.

The structure of our stagewise algorithm is presented in **Algorithm 1**. It can be viewed as a boosting procedure that builds up the solution gradually in terms of forward step (line 4) and backward step (line 3)[3]. The initialization step is solved explicitly (see Lemma 1 below). At each subsequent iteration, the parameter update takes the form $\hat{\mathbf{w}}^{(n)} = \hat{\mathbf{w}}^{(n)} + s\mathbf{1}_{i_n}$ in either forward or backward direction, where $\mathbf{1}_{i_n}$ is a length-$I_n$ vector with all zeros except for a 1 in the $i_n$-th coordinate, $s = \pm\epsilon$, and $\epsilon$ is the pre-specified step size controlling the fineness of the grid. The algorithm also keeps track of the tuning parameter $\lambda$. Intuitively, the selection of the index $(n, i_n)$ and the increment $s$ is guided by minimizing the penalized loss function with the current $\lambda$ subject to a constraint on the step size. Comparing to the standard stagewise LASSO, the main difference here is that we need to select the "best" triple of $(n, i_n, s)$ over all the dimensions across all tensor modes.

Problem (12) and (13) can be solved efficiently. By expansion of $J(\hat{\mathbf{w}}^{(n)} + s\mathbf{1}_{i_n})$, we have

$$J(\hat{\mathbf{w}}^{(n)} + s\mathbf{1}_{i_n}) = \frac{1}{M}(\|\hat{\mathbf{e}}^{(n)}\|_2^2 - 2s\hat{\mathbf{e}}^{(n)\mathrm{T}}\hat{\mathbf{Z}}^{(-n)}\mathbf{1}_{i_n} + \epsilon^2\|\hat{\mathbf{Z}}^{(-n)}\mathbf{1}_{i_n}\|_2^2),$$

where $\widehat{\mathbf{e}}^{(n)} = \widehat{\mathbf{y}} - \widehat{\mathbf{Z}}^{(-n)}\widehat{\mathbf{w}}^{(n)}$ is a constant at each iteration. Then the solution at each forward step is

$$(\widehat{n}, \widehat{i_{\widehat{n}}}) := \underset{n, i_n}{\arg\max}\, 2|\widehat{\mathbf{e}}^{(n)\mathrm{T}}\widehat{\mathbf{Z}}^{(-n)}\mathbf{1}_{i_n}| - \epsilon Diag(\widehat{\mathbf{Z}}^{(-n)\mathrm{T}}\widehat{\mathbf{Z}}^{(-n)})^{\mathrm{T}}\mathbf{1}_{i_n},\; \widehat{s} = sign(\widehat{\mathbf{e}}^{(n)\mathrm{T}}\widehat{\mathbf{Z}}^{(-\widehat{n})}\mathbf{1}_{\widehat{i_{\widehat{n}}}})\epsilon,$$

and at each backward step is

$$(\widehat{n}, \widehat{i_{\widehat{n}}}) := \underset{n, i_n \in I^{(n)}}{\arg\min}\, 2sign(\widehat{w}_{i_n}^{(n)})\widehat{\mathbf{e}}^{\mathrm{T}}\widehat{\mathbf{Z}}^{(-n)}\mathbf{1}_{i_n} + \epsilon Diag(\widehat{\mathbf{Z}}^{(-n)\mathrm{T}}\widehat{\mathbf{Z}}^{(-n)})^{\mathrm{T}}\mathbf{1}_{i_n},$$

where $Diag(\cdot)$ denotes the vector formed by the diagonal elements of a square matrix, and $I^{(n)} \subset \{1, \cdots, I_n\}$ is the $n$-mode active index set at current iteration.

**Computational Analysis**. In Algorithm 1, the most time-consuming part are the calculations of $\widehat{\mathbf{e}}^{(n)\mathrm{T}}\widehat{\mathbf{Z}}^{(-n)}$ and $Diag(\widehat{\mathbf{Z}}^{(-n)\mathrm{T}}\widehat{\mathbf{Z}}^{(-n)})$, involved in both forward and backward steps. We further write as $\widehat{\mathbf{e}}^{(n)\mathrm{T}}\widehat{\mathbf{Z}}^{(-n)} = (\mathbf{Z}^{(-n)}\mathbf{e} - \alpha\beta^{(-n)}M\widehat{\mathbf{w}}^{(n)})^{\mathrm{T}}$, $Diag(\widehat{\mathbf{Z}}^{(-n)\mathrm{T}}\widehat{\mathbf{Z}}^{(-n)}) = Diag(\mathbf{Z}^{(-n)}\mathbf{Z}^{(-n)\mathrm{T}}) + \alpha\beta^{(-n)}M\mathbf{1}^{(n)}$, where $\mathbf{e} = \mathbf{y}^{\mathrm{T}} - \mathbf{Z}^{(-n)\mathrm{T}}\widehat{\mathbf{w}}^{(n)}$ is a constant during each iteration but varies from one iteration to the next, $\mathbf{1}^{(n)}$ is a length-$I_n$ vector with all ones. At each iteration, the computational cost is dominated by the update of $\mathbf{Z}^{(-n)}$ $(n \neq \widehat{n})$, which can be obtained by

$$\mathbf{Z}_{t+1}^{(-n)} = \frac{1}{\sigma_{t+1}}(\sigma_t\mathbf{Z}_t^{(-n)} + \mathbf{Z}_t^{(-n,-\widehat{n})} \times_{\widehat{n}} \widehat{s}_{\widehat{i_{\widehat{n}}}}\mathbf{1}_{\widehat{i_{\widehat{n}}}}), \tag{14}$$

where $(-n, -\widehat{n})$ denotes every mode except $n$ and $\widehat{n}$, which greatly reduces the computation. Therefore, when $n \neq \widehat{n}$, the updates of $\mathbf{Z}^{(-n)}$, $\mathbf{e}$, $\mathbf{Z}^{(-n)}\mathbf{e}$ and $Diag(\mathbf{Z}^{(-n)\mathrm{T}}\mathbf{Z}^{(-n)})$ requires $O(M\prod_{s\neq n,\widehat{n}} I_s + 3MI_n)$, $O(MI_{\widehat{n}})$, $O(MI_n)$, $O(MI_n)$ operations, respectively, when $n = \widehat{n}$, we only need to additionally update $\mathbf{Z}^{(-\widehat{n})}\mathbf{e}$, which requires $O(MI_{\widehat{n}})$ operations. Overall the computational complexity of our approach is $O(M\sum_{n\neq\widehat{n}}^N(\prod_{s\neq n,\widehat{n}} I_s + 5I_n) + 2MI_{\widehat{n}})$ per iteration. In contrast, the ACS algorithm has to be run for each fixed $\lambda$, and within each of such problems it requires $O(M\prod_{n=1}^N I_n)$ per iteration [da Silva et al., 2015].

## 5 Convergence Analysis

We provide convergence analysis for our stagewise algorithm in this section. All detailed proofs are given in the Appendix A. Specifically, Lemma 1 and 2 below justify the validity of the initialization.

**Lemma 1.** *Let $\mathbf{X}$ be the $(N+1)$-mode matricization of $\mathcal{X}$. Denote $\mathbf{X} = [\mathbf{x}_1, \cdots, \mathbf{x}_I]$ where each $\mathbf{x}_i$ is a column of $\mathbf{X}$, then*

$$\lambda_{\max} = 2/M\max\{|\mathbf{x}_i^{\mathrm{T}}\mathbf{y}|; i = 1, \cdots, I.\}.$$

*Moreover, letting $i^* = \arg\max_i|\mathbf{x}_i^{\mathrm{T}}\mathbf{y}|$ and $(i_1^*, \cdots, i_N^*)$ represents its corresponding indices in tensor space, then the initial non-zero solution of (11), denoted as $(\sigma, \{\mathbf{w}^{(n)}\})$, is given by*

$$\sigma = \epsilon, \mathbf{w}^{(1)} = sign(\mathbf{x}_{i^*}^{\mathrm{T}}\mathbf{y})\mathbf{1}_{i_1^*},\; \mathbf{w}^{(n)} = \mathbf{1}_{i_n^*}, \forall n = 2, \cdots, N.$$

*where $\mathbf{1}_{i_n^*}$ is a vector with all 0's except for a 1 in the $i_n^*$-th coordinate.*

**Lemma 2.** *If there exists $s$ and $i_n$ with $|s| = \epsilon, n = 1, \cdots, N$ such that $\Gamma(s\mathbf{1}_{i_1}, \mathbf{1}_{i_2}, \cdots, \mathbf{1}_{i_N}; \lambda) \leq \Gamma(\{\mathbf{0}\}; \lambda)$, it must be true that $\lambda \leq \lambda_0$.*

Lemma 3 shows that the backward step always performs coordinate descent update of fixed size $\epsilon$, each time along the steepest coordinate direction within the current active set, until the descent becomes impossible subject to a tolerance level $\xi$. Also, the forward step performs coordinate descent when $\lambda_{t+1} = \lambda_t$. Lemma 4 shows that when $\lambda$ gets changed, the penalized loss for the previous $\lambda$ can no longer be improved subject to a tolerance level $\xi$. Thus $\epsilon$ controls the granularity of the paths, and $\xi$ is a convergence threshold in optimizing the penalized loss with any fixed tuning parameter. They enable convenient trade-off between computational efficiency and estimation accuracy.

**Lemma 3.** *For any $t$ with $\lambda_{t+1} = \lambda_t$, we have $\Gamma(\sigma_{t+1}, \{\mathbf{w}_{t+1}^{(n)}\}; \lambda_{t+1}) \leq \Gamma(\sigma_t, \{\mathbf{w}_t^{(n)}\}; \lambda_{t+1}) - \xi$.*

**Lemma 4.** *For any $t$ with $\lambda_{t+1} < \lambda_t$, we have $\Gamma(\widehat{\mathbf{w}}_t^{(n)} + s_{i_n}\mathbf{1}_{i_n}; \lambda_t) > \Gamma(\widehat{\mathbf{w}}_t^{(n)}; \lambda_t) - \xi$.*

Lemma 3 and Lemma 4 proves the following convergence theorem.

**Theorem 1.** *For any $t$ such that $\lambda_{t+1} < \lambda_t$, we have $(\sigma_t, \{\mathbf{w}_t^{(n)}\}) \to (\sigma(\lambda_t), \{\widetilde{\mathbf{w}}^{(n)}(\lambda_t)\})$ as $\epsilon, \xi \to 0$, where $(\sigma(\lambda_t), \{\widetilde{\mathbf{w}}^{(n)}(\lambda_t)\})$ denotes a coordinate-wise minimum point of Problem (7).*

Table 1: Compared methods. $\alpha$ and $\lambda$ are regularized parameters; $R$ is the CP rank.

| Methods | LASSO | ENet | Remurs | orTRR | GLTRM | ACS | SURF |
|---|---|---|---|---|---|---|---|
| Input Data Type | Vector | Vector | Tensor | Tensor | Tensor | Tensor | Tensor |
| Regularization | $\ell_1$ ($\mathbf{w}$) | $\ell_1/\ell_2$ ($\mathbf{w}$) | Nuclear/$\ell_1$ ($\mathcal{W}$) | $\ell_2$ ($\mathbf{W}^{(n)}$) | $\ell_1/\ell_2$ ($\mathbf{W}^{(n)}$) | $\ell_1/\ell_2$ ($\mathcal{W}_r$) | $\ell_1/\ell_2$ ($\mathcal{W}_r$) |
| Rank Explored | — | — | — | Optimized | Fixed | Increased | Increased |
| Hyperparameters | $\lambda$ | $\alpha, \lambda$ | $\lambda_1, \lambda_2$ | $\alpha, R$ | $\alpha, \lambda, R$ | $\alpha, \lambda, R$ | $\alpha, \lambda, R$ |

## 6 Experiments

We evaluate the effectiveness and efficiency of our method *SURF* through numerical experiments on both synthetic and real data, and compare with various state-of-the-art regression methods, including *LASSO*, Elastic Net (*ENet*), Regularized multilinear regression and selection (*Remurs*) [Song and Lu, 2017], optimal CP-rank Tensor Ridge Regression (*orTRR*) [Guo et al., 2012], Generalized Linear Tensor Regression Model (*GLTRM*) [Zhou et al., 2013], and a variant of our method with Alternating Convex Search (*ACS*) estimation. Table 1 summarizes the properties of all methods. All methods are implemented in MATLAB and executed on a machine with 3.50GHz CPU and 256GB RAM. For LASSO and ENet we use the MATLAB package glmnet from [Friedman et al., 2010]. For GLTRM, we solve the regularized CP tensor regression simultaneously for all $R$ factors based on TensorReg toolbox [Zhou et al., 2013]. We follow [Kampa et al., 2014] to arrange the test and training sets in the ratio of 1:5. The hyperparameters of all methods are optimized using 5-fold cross validation on the training set, with range $\alpha \in \{0.1, 0.2, \cdots, 1\}$, $\lambda \in \{10^{-3}, 5 \times 10^{-3}, 10^{-2}, 5 \times 10^{-2}, \cdots, 5 \times 10^2, 10^3\}$, and $R \in \{1, 2, \cdots, 50\}$. Specifically, for GLTRM, ACS, and SURF, we simply set $\alpha = 1$. For LASSO, ENet and ACS, we generate a sequence of 100 values for $\lambda$ to cover the whole path. For fairness, the number of iterations for all compared methods are fixed to 100. All cases are run 50 times and the average results on the test set are reported. Our code is available at https://github.com/LifangHe/SURF.

**Synthetic Data**. We first use the synthetic data to examine the performance of our method in different scenarios, with varying step sizes, sparsity level, number of features as well as sample size. We generate the data as follows: $y = \langle \mathbf{X}, \mathbf{W} \rangle + \varepsilon$, where $\varepsilon$ is a random noise generated from $\mathcal{N}(0, 1)$, and $\mathbf{X} \in \mathbb{R}^{I \times I}$. We generate $M$ samples $\{\mathbf{X}^m\}_{m=1}^M$ from $\mathcal{N}(0, \Sigma)$, where $\Sigma$ is a covariance matrix, the correlation coefficient between features $x_{i,j}$ and $x_{p,q}$ is defined as $0.6^{\sqrt{(i-p)^2+(j-q)^2}}$. We generate the true support as $\mathbf{W} = \sum_{r=1}^R \sigma_r \mathbf{w}_r^{(1)} \circ \mathbf{w}_r^{(2)}$, where each $\mathbf{w}_r^{(n)} \in \mathbb{R}^I$, $n = 1, 2$, is a column vector with $\mathcal{N}(0, 1)$ i.i.d. entries and normalized with $\ell_1$ norm, the scalars $\sigma_r$ are defined by $\sigma_r = 1/r$. To impose sparsity on $\mathbf{W}$, we set $S\%$ of its entries (chosen uniformly at random) to zero. When studying one of factors, other factors are fixed to $M = 500$, $I = 16$, $R = 50$, $S = 80$.

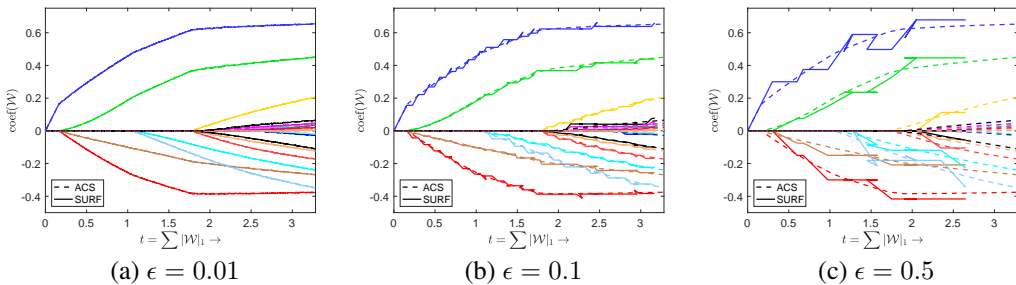

(a) $\epsilon = 0.01$ &emsp;&emsp; (b) $\epsilon = 0.1$ &emsp;&emsp; (c) $\epsilon = 0.5$

Figure 1: Comparison of solution paths of SURF (solid line) and ACS (dashed line) with different step sizes on synthetic data. The path of estimates $\mathcal{W}$ for each $\lambda$ is treated as a function of $t = \|\mathcal{W}\|_1$. Note that when the step size is small, the SURF path is almost indiscernible from the ACS path.

In a first step we analyze the critical parameter $\epsilon$ for our method. This parameter controls how close SURF approximates the ACS paths. Figure 1 shows the solution path plot of our method versus ACS method under both big and small step sizes. As shown by the plots, a smaller step size leads to a closer approximation to the solutions of ACS. In Figure 2, we also provide a plot of averaged prediction error with standard deviation bars (left side of y-axis) and CPU execution time (right side of y-axis in mins) over different values of step size. From the figure, we can see that the choice of the step size affects both computational speed and the root mean-squared prediction error (RMSE). The

smaller the value of step size, the more accurate the regression result but the longer it will take to run. In both figures, the moderate step size $\epsilon = 0.1$ seems to offer a better trade-off between performance and ease of implementation. In the following we fix $\epsilon = 0.1$.

Next, we examine the performance of our method with varying sparsity level of $\mathbf{W}$. For this purpose, we compare the prediction error (RMSE) and running time (log min) of all methods on the synthetic data. Figure 3(a)-(b) shows the results for the case of $S = \{60, 80, 90, 95, 98\}$ on synthetic 2D data, where $S\%$ indicates the sparsity level of true $\mathbf{W}$. As can be seen from the plots, SURF generates slightly better predictions than other existing methods when the true $\mathbf{W}$ is sparser. Moreover, as shown in Figure 3(c), it is also interesting to note that larger step sizes give much more sparsity of coefficients for SURF, this explains why there is no value for some curves as the increase of $\sum |\mathcal{W}|_1$ in Figure 1(c).

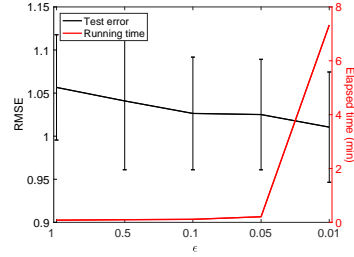

Figure 2: Results to different values of step size $\epsilon$.

Furthermore, we compare the prediction error and running time (log min) of all methods with increasing number of features. Figure 4(a)-(b) shows the results for the case of $I = \{8, 16, 32, 64\}$ on synthetic 2D data. Overall, SURF gives better predictions at a lower computational cost. Particularly, SURF and ACS have very similar prediction qualities, this matches with our theoretical result on the solutions of SURF versus ACS. SURF achieves better predictions than other tensor methods, indicating the effectiveness of structured sparsity in unit-rank tensor decomposition itself. In terms of running time, it is clear that as the number of features is increased, SURF is significantly faster than other methods.

Finally, Figure 5 shows the results with increasing number of samples. Overall, SURF gives better predictions at a lower computational cost. Particularly, Remurs and orTRR do not change too much as increasing number of samples, this may due to early stop in the iteration process when searching for optimized solution.

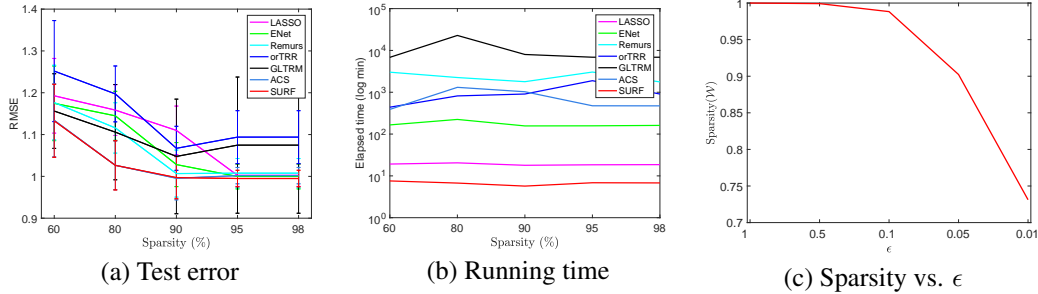

| (a) Test error | (b) Running time | (c) Sparsity vs. $\epsilon$ |

Figure 3: Results with increasing sparsity level ($S\%$) of true $\mathbf{W}$ on synthetic 2D data (a)-(b), and (c) sparsity results of $\mathcal{W}$ versus step size for SURF.

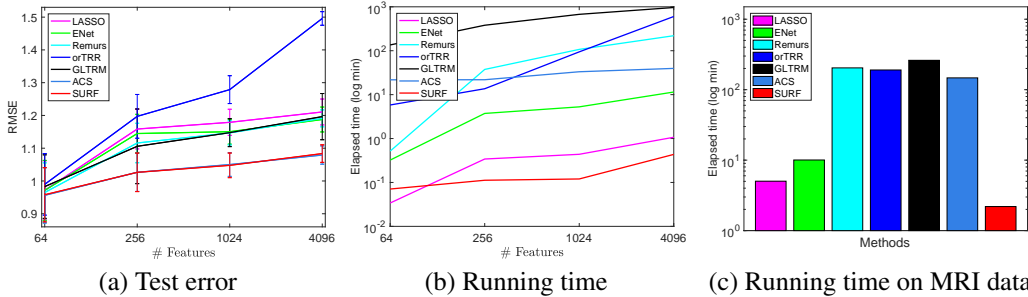

| (a) Test error | (b) Running time | (c) Running time on MRI data |

Figure 4: Results with increasing number of features on synthetic 2D data (a)-(b), and (c) real 3D MRI data of features $240 \times 175 \times 176$ with fixed hyperparameters (without cross validation).

**Real Data**. We also examine the performance of our method on a real medical image dataset, including both DTI and MRI images, obtained from the Parkinson's progression markers initiative

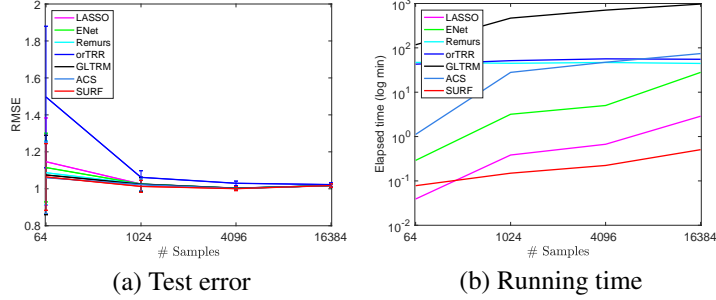

(a) Test error          (b) Running time

Figure 5: Results with increasing number of samples on synthetic 2D data.

Table 2: Performance comparison over different DTI datasets. Column 2 indicates the used metrics RMSE, Sparsity of Coefficients (SC) and CPU execution time (in mins). The results are averaged over 50 random trials, with both the mean values and standard deviations (mean $\pm$ std.)

| Datasets | Metrics | Comparative Methods | | | | | | |
|---|---|---|---|---|---|---|---|---|
| | | LASSO | ENet | Remurs | orTRR | GLTRM | ACS | SURF |
| $DTI_{fact}$ | RMSE | 2.94±0.34 | 2.92±0.32 | 2.91±0.32 | 3.48±0.21 | 3.09±0.35 | 2.81±0.24 | 2.81±0.23 |
| | Sparsity | 0.99±0.01 | 0.97±0.01 | 0.66±0.13 | 0.00±0.00 | 0.90±0.10 | 0.92±0.02 | 0.95±0.01 |
| | Time | 6.4±0.3 | 46.6±4.6 | 161.3±9.3 | 27.9±5.6 | 874.8±29.6 | 60.8±24.4 | 1.7±0.2 |
| $DTI_{rk2}$ | RMSE | 3.18±0.36 | 3.16±0.42 | 2.97±0.30 | 3.76±0.44 | 3.26±0.46 | 2.90±0.31 | 2.91±0.32 |
| | Sparsity | 0.99±0.01 | 0.95±0.03 | 0.37±0.09 | 0.00±0.00 | 0.91±0.06 | 0.93±0.02 | 0.94±0.01 |
| | Time | 5.7±0.3 | 42.4±2.9 | 155.0±10.7 | 10.2±0.1 | 857.4±22.5 | 63.0±21.6 | 5.2±0.8 |
| $DTI_{sl}$ | RMSE | 3.06±0.34 | 2.99±0.34 | 2.93±0.27 | 3.56±0.41 | 3.14±0.39 | 2.89±0.38 | 2.87±0.35 |
| | Sparsity | 0.98±0.01 | 0.95±0.01 | 0.43±0.17 | 0.00±0.00 | 0.87±0.03 | 0.90±0.03 | 0.93±0.02 |
| | Time | 5.8±0.3 | 45.0±1.0 | 163.6±9.0 | 7.5±0.9 | 815.4±6.5 | 66.3±44.9 | 1.5±0.1 |
| $DTI_{tl}$ | RMSE | 3.20±0.40 | 3.21±0.59 | 2.84±0.35 | 3.66±0.35 | 3.12±0.32 | 2.82±0.33 | 2.83±0.32 |
| | Sparsity | 0.99±0.01 | 0.96±0.03 | 0.44±0.13 | 0.00±0.00 | 0.86±0.03 | 0.90±0.02 | 0.91±0.02 |
| | Time | 5.5±0.2 | 42.3±1.4 | 159.6±7.6 | 26.6±3.1 | 835.8±9.9 | 96.7±43.2 | 3.8±0.5 |
| Combined | RMSE | 3.02±0.37 | 2.89±0.41 | 2.81±0.31 | 3.33±0.27 | 3.26±0.45 | 2.79±0.31 | 2.78±0.29 |
| | Sparsity | 0.99±0.00 | 0.97±0.01 | 0.34±0.22 | 0.00±0.00 | 0.91±0.19 | 0.97±0.01 | 0.99±0.00 |
| | Time | 8.8±0.6 | 71.9±2.3 | 443.5±235.7 | 48.4±8.0 | 1093.6±49.7 | 463.2±268.1 | 6.2±0.5 |

(PPMI) database[4] with 656 human subjects. We parcel the brain into 84 regions and extract four types connectivity matrices. Our goal is to predict the Montreal Cognitive Assessment (MoCA) scores for each subject. Details of data processing are presented in Appendix B. We use three metrics to evaluate the performance: root mean squared prediction error (RMSE), which describes the deviation between the ground truth of the response and the predicted values in out-of-sample testing; sparsity of coefficients (SC), which is the same as the $S\%$ defined in the synthetic data analysis (i.e., the percentage of zero entries in the corresponding coefficient); and CPU execution time. Table 2 shows the results of all methods on both individual and combined datasets. Again, we can observe that SURF gives better predictions at a lower computational cost, as well as good sparsity. In particular, the paired t-tests showed that for all five real datasets, the RMSE and SC of our approach are significantly lower and higher than those of Remurs and GLTRM methods, respectively. This indicates that the performance gain of our approach over the other low-rank + sparse methods is indeed significant. Figure 4(c) provides the running time (log min) of all methods on the PPMI MRI images of $240 \times 175 \times 176$ voxels each, which clearly demonstrates the efficiency of our approach.

## Acknowledgements

This work is supported by NSF No. IIS-1716432 (Wang), IIS-1750326 (Wang), IIS-1718798 (Chen), DMS-1613295 (Chen), IIS-1749940 (Zhou), IIS-1615597 (Zhou), ONR N00014-18-1-2585 (Wang), and N00014-17-1-2265 (Zhou), and Michael J. Fox Foundation grant number 14858 (Wang). Lifang He's research is supported in part by 1R01AI130460. Data used in the preparation of this article were obtained from the Parkinson's Progression Markers Initiative (PPMI) database (http://www.ppmi-info.org/data). For up-to-date information on the study, visit http://www.ppmi-info.org. PPMI – a public-private partnership – is funded by the Michael J. Fox Foundation for Parkinson's Research and funding partners, including Abbvie, Avid, Biogen, Bristol-Mayers Squibb, Covance, GE, Genentech, GlaxoSmithKline, Lilly, Lundbeck, Merk, Meso Scale Discovery, Pfizer, Piramal, Roche, Sanofi, Servier, TEVA, UCB and Golub Capital.

## Footnotes

[2] As stated in [Zhao and Yu, 2007], $\xi$ is implementation-specific but not necessarily a user parameter. In all experiments, we set $\xi = \epsilon^2/2$.

[3] Boosting amounts to combine a set of "weak learners" to build one strong learner, and is connected to stagewise and regularized estimation methods. SURF is a boosting method since each backward/forward step is in essence finding a weaker learner to incrementally improve the current learner (thus generate a path of solutions).

[4] http://www.ppmi-info.org/data

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
