[Supplementary Material]

# Supplementary Material for the Paper: Boosted Sparse and Low-Rank Tensor Regression

**Lifang He**
Weill Cornell Medicine
lifanghescut@gmail.com

**Kun Chen**[*]
University of Connecticut
kun.chen@uconn.edu

**Wanwan Xu**
University of Connecticut
wanwan.xu@uconn.edu

**Jiayu Zhou**
Michigan State Universtiy
dearjiayu@gmail.com

**Fei Wang**
Weill Cornell Medicine
few2001@med.cornell.edu

## A  Proof of Theorem and Lemmas

**Lemma 1.** *Let* $\mathbf{X}$ *be the* $(N+1)$*-mode matricization of* $\mathcal{X}$*. Denote* $\mathbf{X} = [\mathbf{x}_1, \cdots, \mathbf{x}_I]$ *where each* $\mathbf{x}_i$ *is a column of* $\mathbf{X}$*, then*

$$\lambda_{\max} = 2/M \max\{|\mathbf{x}_i^{\mathrm{T}}\mathbf{y}|; i = 1, \cdots, I.\}.$$

*Moreover, letting* $i^* = \arg\max_i |\mathbf{x}_i^{\mathrm{T}}\mathbf{y}|$ *and* $(i_1^*, \cdots, i_N^*)$ *represents its corresponding indices in tensor space, then the initial non-zero solution of (11), denoted as* $(\sigma, \{\mathbf{w}^{(n)}\})$*, is given by*

$$\sigma = \epsilon, \mathbf{w}^{(1)} = sign(\mathbf{x}_{i^*}^{\mathrm{T}}\mathbf{y})\mathbf{1}_{i_1^*}, \ \mathbf{w}^{(n)} = \mathbf{1}_{i_n^*}, \forall n = 2, \cdots, N.$$

*where* $\mathbf{1}_{i_n^*}$ *is a vector with all 0's except for a 1 in the* $i_n^*$*-th coordinate.*

*Proof.* By using multilinear algebra, the problem (8) can be equivalently written as

$$\min_{\{\sigma, w^{(n)}\}} \frac{1}{M} \|\mathbf{y} - \mathbf{X}(\sigma\mathbf{w}^{(N)} \otimes \cdots \otimes \mathbf{w}^{(1)}\|_2^2 + \lambda\sigma \prod_{n=1}^{N}\|\mathbf{w}^{(n)}\|_1 + \alpha\sigma^2 \prod_{n=1}^{N}\|\mathbf{w}^{(n)}\|_2^2$$

$$\text{s.t.} \quad \sigma \geq 0, \ \|\mathbf{w}^{(n)}\|_1 = 1, \ n = 1, \cdots, N. \tag{1}$$

where $\otimes$ denotes the Kronecker product operator.

This problem has the same $\lambda_{\max}$ as its corresponding elastic net problem by considering $(\sigma\mathbf{w}^{(N)} \otimes \cdots \otimes \mathbf{w}^{(1)})$ as a whole. Thus $\lambda_{\max}$ and the initial non-zero solution can be obtained as above by the Karush-Kuhn-Tucker (KKT) optimality conditions for the elastic net problem. $\square$

**Lemma 2.** *If there exists* $s$ *and* $i_n$ *with* $|s| = \epsilon, n = 1, \cdots, N$ *such that*

$$\Gamma(s\mathbf{1}_{i_1}, \mathbf{1}_{i_2}, \cdots, \mathbf{1}_{i_N}; \lambda) \leq \Gamma(\{\mathbf{0}\}; \lambda), \tag{2}$$

*it must be true that* $\lambda \leq \lambda_0$*.*

*Proof.* By assumption, we can expand (2) as

$$J(s\mathbf{1}_{i_1}, \mathbf{1}_{i_2}, \cdots, \mathbf{1}_{i_N}) + \lambda\Omega(s\mathbf{1}_{i_1}, \mathbf{1}_{i_2}, \cdots, \mathbf{1}_{i_N}) \leq J(\{\mathbf{0}\}).$$

It follows that

$$\lambda \leq \frac{1}{\epsilon}(J(\{\mathbf{0}\}) - J(s\mathbf{1}_{i_1}, \mathbf{1}_{i_2}, \cdots, \mathbf{1}_{i_N}))$$

$$\leq \frac{1}{\epsilon}(J(\{\mathbf{0}\}) - \min_{\{i_1, \cdots, i_N\}, s = \pm\epsilon} J(s\mathbf{1}_{i_1}, \mathbf{1}_{i_2}, \cdots, \mathbf{1}_{i_N}))$$

$$= \lambda_0. \qquad \square$$

---

[*]Corresponding Author

**Lemma 3.** *For any $t$ with $\lambda_{t+1} = \lambda_t$, we have $\Gamma(\sigma_{t+1}, \{\mathbf{w}_{t+1}^{(n)}\}; \lambda_{t+1}) \leq \Gamma(\sigma_t, \{\mathbf{w}_t^{(n)}\}; \lambda_{t+1}) - \xi$.*

*Proof.* This is obviously true if the backward step is taken since $\Gamma(\sigma_{t+1}, \{\mathbf{w}_{t+1}^{(n)}\}; \lambda_t) \leq \Gamma(\sigma_t, \{\mathbf{w}_t^{(n)}\}; \lambda_t) - \xi$ and $\lambda_{t+1} = \lambda_t$. So we only need to consider the forward step when $\lambda_{t+1} = \lambda_t$. If the claim is not true, then

$$J(\sigma_t, \{\mathbf{w}_t^{(n)}\}) - J(\sigma_{t+1}, \{\mathbf{w}_{t+1}^{(n)}\}) < \lambda_t \Omega(\sigma_{t+1}, \{\mathbf{w}_{t+1}^{(n)}\}) - \lambda_t \Omega(\sigma_t, \{\mathbf{w}_t^{(n)}\}) + \xi = \lambda_t \epsilon + \xi.$$

That is,

$$\lambda_{t+1} = \lambda_t > \frac{1}{\epsilon}(J(\sigma_t, \{\mathbf{w}_t^{(n)}\}) - J(\sigma_{t+1}, \{\mathbf{w}_{t+1}^{(n)}\}) - \xi),$$

which contradicts with the fact that $\lambda_{t+1} = \min(\lambda_t, \frac{1}{\epsilon}(J(\sigma_t, \{\mathbf{w}_t^{(n)}\}) - J(\sigma_{t+1}, \{\mathbf{w}_{t+1}^{(n)}\}) - \xi))$. $\square$

**Lemma 4.** *For any $t$ with $\lambda_{t+1} < \lambda_t$, we have $\Gamma(\hat{\mathbf{w}}_t^{(n)} + s_{i_n} \mathbf{1}_{i_n}; \lambda_t) > \Gamma(\hat{\mathbf{w}}_t^{(n)}; \lambda_t) - \xi$.*

*Proof.* First of all, when $\lambda_{t+1} < \lambda_t$, it holds that $\Omega(\sigma_{t+1}, \{\mathbf{w}_{t+1}^{(n)}\}) = \Omega(\sigma_t, \{\mathbf{w}_t^{(n)}\}) + \epsilon$. From $\lambda_{t+1} = \min(\lambda_t, \frac{1}{\epsilon}(J(\sigma_t, \{\mathbf{w}_t^{(n)}\}) - J(\sigma_{t+1}, \{\mathbf{w}_{t+1}^{(n)}\}) - \xi))$ and $\lambda_{t+1} < \lambda_t$, we know that

$$J(\sigma_t, \{\mathbf{w}_t^{(n)}\}) - J(\sigma_{t+1}, \{\mathbf{w}_{t+1}^{(n)}\}) - \xi = \lambda_{t+1}\epsilon = \lambda_{t+1}(\Omega(\sigma_{t+1}, \{\mathbf{w}_{t+1}^{(n)}\}) - \Omega(\sigma_t, \{\mathbf{w}_t^{(n)}\})),$$

that is, $\Gamma(\hat{\mathbf{w}}_t^{(n)}; \lambda_{t+1}) - \xi = \Gamma(\hat{\mathbf{w}}_{t+1}^{(n)}; \lambda_{t+1})$. Then we have

$$\begin{aligned}
\Gamma(\hat{\mathbf{w}}_t^{(n)}; \lambda_t) - \xi &= \Gamma(\hat{\mathbf{w}}_t^{(n)}; \lambda_{t+1}) - \xi + (\lambda_t - \lambda_{t+1})\Omega(\sigma_t, \{\mathbf{w}_t^{(n)}\}) \\
&= \Gamma(\hat{\mathbf{w}}_{t+1}^{(n)}; \lambda_{t+1}) + (\lambda_t - \lambda_{t+1})\Omega(\sigma_t, \{\mathbf{w}_t^{(n)}\}) \\
&= \Gamma(\hat{\mathbf{w}}_{t+1}^{(n)}; \lambda_t) + (\lambda_{t+1} - \lambda_t)(\Omega(\sigma_{t+1}, \{\mathbf{w}_{t+1}^{(n)}\}) - \Omega(\sigma_t, \{\mathbf{w}_t^{(n)}\})) \\
&= \Gamma(\hat{\mathbf{w}}_{t+1}^{(n)}; \lambda_t) + (\lambda_{t+1} - \lambda_t)\epsilon < \Gamma(\hat{\mathbf{w}}_{t+1}^{(n)}; \lambda_t) = \min\{\Gamma(\hat{\mathbf{w}}_t^{(n)} + s_{i_n}\mathbf{1}_{i_n}; \lambda_t)\}. \square
\end{aligned}$$

**Theorem 1.** *For any $t$ such that $\lambda_{t+1} < \lambda_t$, we have $(\sigma_t, \{\mathbf{w}_t^{(n)}\}) \to (\sigma(\lambda_t), \{\widetilde{\mathbf{w}}^{(n)}(\lambda_t)\})$ as $\epsilon, \xi \to 0$, where $(\sigma(\lambda_t), \{\widetilde{\mathbf{w}}^{(n)}(\lambda_t)\})$ denotes a coordinate-wise minimum point of Problem (7).*

*Proof.* First, by Lemma 3, we have $\Gamma(\sigma_t, \{\mathbf{w}_t^{(n)}\}; \lambda_t) \leq \Gamma(\sigma_{t-1}, \{\mathbf{w}_{t-1}^{(n)}\}; \lambda_{t-1}) - \xi$ when $\lambda_t = \lambda_{t-1}$. Then it is easy to verify the series of inequalities

$$\Gamma(\sigma_t, \{\mathbf{w}_t^{(n)}\}; \lambda_t) \leq \Gamma(\sigma_{t-1}, \{\mathbf{w}_{t-1}^{(n)}\}; \lambda_{t-1}) - \xi \leq \cdots \leq \Gamma(\sigma_{t-p}, \{\mathbf{w}_{t-p}^{(n)}\}; \lambda_{t-p}) - p\xi \quad (3)$$

holds when $\lambda_t = \lambda_{t-1} = \cdots = \lambda_{t-p}$ and $p$ is the value such that $\lambda_{t-p} < \lambda_{t-p-1}$. As $\epsilon, \xi \to 0$, a straightforward consequence of (3) is that the sequence of the objective function values is monotonically decreasing at $\lambda_t$, that is,

$$\Gamma(\sigma_t, \{\mathbf{w}_t^{(n)}\}; \lambda_t) \leq \Gamma(\sigma_{t-1}, \{\mathbf{w}_{t-1}^{(n)}\}; \lambda_t) \leq \cdots \leq \Gamma(\sigma_{t-p}, \{\mathbf{w}_{t-p}^{(n)}\}; \lambda_t). \quad (4)$$

Using Lemma 4, we know that $\lambda_t$ gets reduced such that $\lambda_{t+1} < \lambda_t$ only occurs in the forward step when $\Gamma(\sigma_{t+1}, \{\mathbf{w}_{t+1}^{(n)}\}; \lambda_t) > \Gamma(\sigma_t, \{\mathbf{w}_t^{(n)}\}; \lambda_t) - \xi$. This means that even by searching over all possible coordinate descent directions in each subproblem (with the size of update fixed at $\epsilon$), the objective function at $\lambda_t$ can not be further reduced. Since each subproblem is strongly convex w.r.t $(\sigma, \mathbf{w}^{(n)})$, it has a unique solution. Therefore, when $\epsilon, \xi \to 0$ and at the time $\lambda_t$ gets reduced to $\lambda_{t+1}$, we can say a coordinate-wise minimum point of $\Gamma(\cdot)$ is reached for $\lambda_t$, which completes the proof. $\square$

## B  Description of Data Preprocessing

We preprocessed the DTI and MRI acquisitions on 656 subjects as follows. T1-weighted MRI data was acquired using the ADNI-2 sequence, and processed using the FreeSurfer[2], followed by [1]. For DTI data, each subject's raw data were aligned to the b0 image using the FSL[3] eddy-correct tool to correct for head motion and eddy current distortions. The gradient table is also corrected accordingly. Non-brain tissue is removed from the diffusion MRI using the Brain Extraction Tool (BET) from FSL [2]. To correct for echo-planar induced (EPI) susceptibility artifacts, which can cause distortions at tissue-fluid interfaces, skull-stripped b0 images are linearly aligned and then elastically registered to their respective preprocessed structural MRI using Advanced Normalization Tools (ANTs[4]) with

SyN nonlinear registration algorithm [3]. The resulting 3D deformation fields are then applied to the remaining diffusion-weighted volumes to generate full preprocessed diffusion MRI dataset for the brain network reconstruction. In the meantime, 84 ROIs is parcellated from T1-weighted MRI using Freesufer.

Based on these 84 ROIs, we reconstruct four types of brain connectivity matrices for each subject, using the following four tensor-based deterministic tractography algorithms: Fiber Assignment by Continuous Tracking (FACT) [4], the 2nd-order Runge-Kutta (RK2) [5], interpolated streamline (SL) [6], and the tensorline (TL) [7]. Each resulted connectivity matrix for each subject is $84 \times 84$. To avoid computation bias, we normalize each connectivity matrix by dividing by its maximum value, as matrices derived from different tractography methods have different scales and ranges.

## Footnotes

[2] https://surfer.nmr.mgh.harvard.edu

[3] http://www.fmrib.ox.ac.uk/fsl

[4] http://stnava.github.io/ANTs/