[Reviews · NeurIPS 2018]

Reviewer 1



In this paper, a novel sparse and low-rank tensor regression model along with a fast solver are developed and analysed. Concretely, a parsimonious and interpretable mapping is learnt by seeking a sparse CP decomposition of coefficient tensor. To solve this problem an (asymptotically) convergent solver is developed. The performance of the proposed algorithm is thoroughly evaluated by conducting numerical tests in both real and synthetic data. The reported results indicate that the proposed method is faster and in several cases achieves better predictions than the state of the art when comparable RMSE and sparsity are achieved. Overall, the paper is well-written, and easy to read. Appropriate motivations are clearly stated in the introduction of the paper. I believe that the current version of the paper can be published.

Reviewer 2



This paper examines the problem of tensor regression and proposes a boosted sparse low-rank model that produces interpretable results. In their low-rank tensor regression model, unit-rank tensors from the CP decomposition of the coefficient tensor is assumed to be sparse. This assumption allows for an interpretable model where the outcome is related to only a subset of features. For model estimation, the authors use a divide-and-conquer strategy to learn the sparse CP decomposition, based on an existing sequential extraction method, where sparse unit-rank problems are sequentially solved. Instead of using an alternating convex search (ACS) approach, the authors use a stage-wise unit-rank tensor factorization algorithm to learn the model. A convergence analysis of the stage-wise algorithm is provided. The novel contributions of this paper appear to be the the imposition of sparsity on the CP decomposition of the low-rank tensor regression model and the boosting unit-rank tensor factorization algorithm. Experimentally, the paper demonstrates that the boosting algorithm performs equivalently as an alternating convex search approach and that the boosting algorithm appears to be advantageous in performance time only when the number of features is high. This is further illustrated on a medial data set, where the boosting + ACS approaches both perform similarity in terms of predictive performance + sparsity (with ACS requiring more computational time); comparing the proposed approach to other low-rank tensor methods + sparse regression methods, it appears that there is not always a significant improvement offered in predictive performance + sparsity. If there is one clear benefit offered by the proposed boosting, it is faster computational time. I read the authors' feedback and thank them for their comments. The t-tests to check for significance in performance on the 5 experimental data sets are helpful to take into consideration. I also do appreciate the big-O computational complexity analysis per iteration.

Reviewer 3



The paper considers a low-rank tensor regression model with sparsity, i.e. some entries of parameter tensors are zero. Specifically, the authors apply a divide-and-conquer strategy to make the estimation procedure more efficient and develop a stagewise estimation procedure to efficiently approximate the solution path. In general, the paper is well written and the results of experiments are convincing. Some comments are listed as follows. 1. In the Section ‘Experiments’’, especially for the synthetic data, the paper only reports prediction error and running time. What about the ability of SURF in terms of ‘variable selection’? This is important for model interpretability. 2. What is the definition of ‘sparsity of coefficients’ in the real data analysis?

Reviewer 4



In this paper, the authors propose an algorithm for tensor regression. The problem of tensor regression with low-rank models have been approached in several recent works. The novelty in the present paper is that, besides the low-rank CP (PARAFAC) property, sparsity in the factor matrices are also imposed, thus providing interpretability to the results. In order to estimate the CP model factors with sparsity constraints, the authors propose a deflation approach which means to estimate one rank-1 term of the decomposition at a time, giving that the other terms are fixed. For the rank-1 term estimation step, the authors use a stagewise technique inspired in the algorithm for Lasso (Zhow & Yu, 2007). Finally, the experimental results are presented using simulated as well as real-world datasets. I found the paper somewhat relevant to NIPS, although its contribution can be seen as an improvement of previously proposed regression techniques. Particularly, I found the following issues in the current version of the paper: - The authors made use of the term “boosted” in the title but it is not well justified in the paper. In particular, the explanation in lines 158 – 160 is not clear. Does it mean that using a combination of a forward plus a backward step provides a boosting effect? The Authors have clarified this in their responses. - Line 98: The sentence “3) It leads to a more flexible and parsimonious model, thus makes the model generalizability better” is kind of vague and should be more precise. What does it mean to have a more flexible model? What does it mean to have a more parsimonious model? What do the authors mean by better generalizability? Are the authors referring to the fact that the model is less prone to overfitting? If so, this must be clearly stated and demonstrated, at least, with experiments. The Authors have clarified this in their responses. - Line 114-115: The sentence “Here for improving the convexity of the problem and its numerical stability …” is not clear enough. These concepts must be discussed more precisely. Is the model not convex without using the elastic-net? What does it mean to be numerically unstable and why the elastic-net model helps to avoid it?The Authors have clarified this in their responses and will provide further clarification in the final version. - Line 116 – 117: The authors say that “we fix alpha>0 as a small constant”. What does it mean a "small value"? What is the range of this parameter and how should we judge if a value is small or large? The Authors have clarified this in their responses and will make more specific recommendations on the choice of alpha in the final version of the paper. - Algorithm 1: This algorithm is called “Fast” by the authors. The authors should provide some complexity analysis to justify the term “fast” in the algorithm name. - I found the experimental results based on real-world signals somewhat confusing. The authors state that the goal in this case is to predict the Montreal Cognitive Assessment scores. The authors evaluate the results in terms of root mean squares error (RMSE), sparsity and computation time. This error (RMSE) is not defined in the paper. Is it computed by calculating the error between the ground truth score and the one predicted by the model? The Authors have clarified this in their responses and will provide a definition in the final version of the paper. - Line 66, the index in the lower end in the last sum must be changed i_1 -> i_N - Line 35 in supplemental information: “Some Experimental Results” -> “Additional Experimental Results”